# LLMs learn governing principles of dynamical systems, revealing an in-context neural scaling law

**Toni J.B. Liu** [1] **Nicolas Boullé** [2] **Raphaël Sarfati** [3] **Christopher J. Earls** [3][4]

## Abstract

Pretrained large language models (LLMs) are surprisingly effective at performing zero-shot tasks, including time-series forecasting. However, understanding the mechanisms behind such capabilities remains highly challenging due to the complexity of the models. We study LLMs' ability to extrapolate the behavior of dynamical systems whose evolution is governed by principles of physical interest. Our results show that LLaMA 2, a language model trained primarily on texts, achieves accurate predictions of dynamical system time series without fine-tuning or prompt engineering. Moreover, the accuracy of the learned physical rules increases with the length of the input context window, revealing an in-context version of neural scaling law. Along the way, we present a flexible and efficient algorithm for extracting probability density functions of multi-digit numbers directly from LLMs. The code and data supporting this study are available at: https://github.com/AntonioLiu97/llmICL.

## 1. Introduction

Since the introduction of the transformer architecture (Vaswani et al., 2017), Large language models (LLMs) have shown a variety of unexpected emergent properties, such as program execution (Nye et al., 2021) and multi-step reasoning (Cobbe et al., 2021; Wei et al., 2022; Suzgun et al., 2022).

Despite the empirical success of LLMs, how they compress vast amounts of information and implement complex algorithms within their architecture is not readily discoverable. To this end, recent studies used representation probes to decipher how concepts and functions are encoded in the layers of trained neural networks (Akyürek et al., 2022; Gurnee & Tegmark, 2023; Marks & Tegmark, 2023; Park et al., 2023; Hendel et al., 2023).

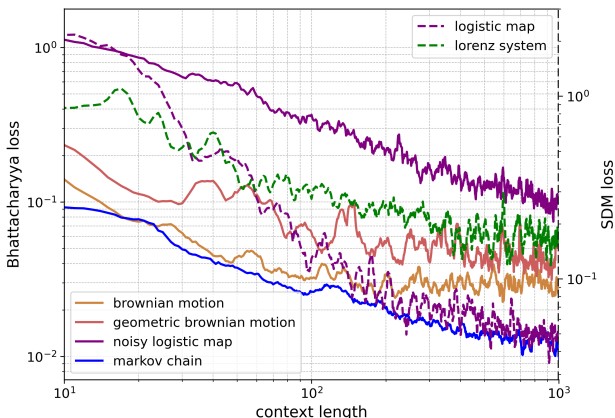

*Figure 1.* Evolution of the loss function for the predicted next state by LLaMA-13b with respect to the number of observed states in various physical systems. We employ the Bhattacharyya distance as a loss function for stochastic systems (solid lines), and the squared deviations from the mean (SDM) for deterministic systems (dashed lines). Brownian motion and geometric Brownian motion deviate significantly from power law scaling, which can be explained by their lack of stationary distributions (Appendix A.8).

Our work explores LLMs' ability to model the world by proposing a new perspective and empirical approach. Inspired by the recent observations that LLMs are capable of in-context time series extrapolation without specific prompting or fine-tuning (Gruver et al., 2023; Jin et al., 2023a), we aim to quantify LLMs' ability to extrapolate stochastic dynamical systems. We find that, as the number of observed time steps increases, an LLM's statistical prediction consistently converges to the ground truth transition rules underlying the system; leading to an empirical scaling law, as observed in Figure 1.

Our **main contributions** are as follows:

[1]Department of Physics, Cornell University, Ithaca, NY, 14853, USA [2]Department of Applied Mathematics and Theoretical Physics, University of Cambridge, Cambridge, CB3 0WA, UK [3]School of Civil and Environmental Engineering, Cornell University, Ithaca, NY, 14853, USA [4]Center for Applied Mathematics, Cornell University, Ithaca, NY, 14853, USA. Correspondence to: Christopher J. Earls <earls@cornell.edu>.

*Proceedings of the 1ˢᵗ Workshop on In-Context Learning at the 41ˢᵗ International Conference on Machine Learning*, Vienna, Austria. 2024. Copyright 2024 by the author(s).

- demonstrating LLMs' zero-shot ability to model the evolution of dynamical systems without instruction prompting;
- implementing a computationally efficient framework called *Hierarchy-PDF* to extract statistical information of a dynamical system learned by a transformer-based LLM;
- discovering a scaling law between the accuracy in the learned transition rules (compared to ground truth) and the context window length.

## 2. Background and related work

*In-context learning* refers to LLM's emergent ability to learn from examples included in the prompt (Brown et al., 2020). One example of in-context learning is zero-shot time series forecasting (Gruver et al., 2023).

The work of (Gruver et al., 2023) aims to forecast empirical time series and introduces a tokenization procedure to convert a sequence of floating point numbers into appropriate textual prompts for LLMs. This led to several subsequent studies on the application of LLMs for time series forecasting (Chen et al., 2023; Jin et al., 2023b;a; Dooley et al., 2023; Schoenegger & Park, 2023; Wang et al., 2023; Xu et al., 2023).

Unlike these prior studies, our work does not focus on forecasting real-world time series, such as weather data or electricity demand, where the underlying model generating the sequence is unavailable or undefined. Instead, we aim to *extract the learned transition rules from the probability vector generated by the LLM* and compare them against the ground truth rules (chaotic, stochastic, discrete, continuous, etc.) governing the input time series.

## 3. Methodology

Our methodology for testing LLMs' ability to learn physical rules from in-context data consists of three steps:
1. Sample a time series $\{x_t\}_{t \geq 0}$ from a given dynamical system governed by Markovian transition rules $P_{ij}$.
2. Prompt the LLM with this time series to extract the learned probability densities for subsequent digits $\tilde{P}_{ij}$.
3. Measure the discrepancy, between the ground truth $P_{ij}$ and learned $\tilde{P}_{ij}$, using Bhattacharyya distance. [1]

### 3.1. Prompt generation

**Markov processes.** Most of our testing data may be modeled as discrete-time Markov chains, where the probability distribution function (PDF) of the next state at time $t + 1$ depends solely on the previous state $x_t$ at time $t$:

$$P(X_{t+1}|x_1, \ldots, x_t) = P(X_{t+1}|x_t).$$

---

[1] Other loss functions may be appropriate depending on whether the dynamical system is stochastic or deterministic, see Appendix A.1

This models either discrete iterative systems or continuous dynamical systems after time-discretization.

**Time series tokenization.** An input time series typically consists of ($\sim 10^3$) time steps, each represented as a real number. We first rescale each number and represent it using $n$ digits (typically, $n = 3$). Each time series is rescaled to the interval $[1.50, 8.50]$ so that the number of digits never changes throughout the series. We then follow the scheme introduced in (Gruver et al., 2023) to serialize the time series as strings and tokenize them.

### 3.2. Extraction of transition rules

**Discrete state space.** When the Markov process is discrete and has a finite state space, each state can be represented by a single token. We employ tokens corresponding to the ten number strings: $0, \ldots, 9$. We find that even the most sophisticated LLaMA model (LLaMA-70b) can only learn up to 9 discrete states. Therefore, we do not attempt to go beyond 9 distinct states by extending to non-number tokens (see Appendix A.3.3 for more details).

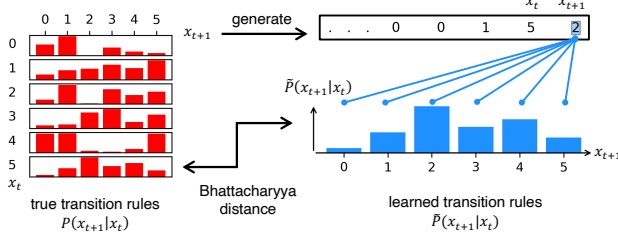

*Figure 2.* Extracting learned transition rules of systems with discrete state space.

Figure 2 illustrates our framework for learning discrete Markov chains with LLMs. First, we randomly sample an $n \times n$ transition matrix $(P_{ij})$. We then generate a Markov chain according to $P_{ij}$, tokenize the time series and pass to an LLM with no additional "prompt engineering". The length of the series is chosen such that the tokenized representation does not exceed the length of the LLM's context window. We extract the LLM's prediction for the next state by performing a softmax operation on the output logits corresponding to the $n$ allowed states and discarding all other logits.

**Continuous state space.** Many stochastic processes, such as the Brownian motion (Einstein, 1905; Perrin, 1909), are supported on continuous state space. For these processes, we represent the value of each state as a multi-digit number and separate each state using the comma symbol ",". As observed in (Jin et al., 2023a), an LLM prediction of multi-digit values can be naturally interpreted as a hierarchical softmax distribution (Mnih & Hinton, 2008; Challu et al., 2023).

Specifically, let $u$ denote a multi-digit string representing the value of a state at a given time-step, then the LLM's softmax prediction for the $i^{th}$ digit, $u_i$, provides a histogram of ten bins of width $0.1^i$. Subsequently, the prediction of the $(i+1)^{\text{th}}$ digit goes down one level into the hierarchical tree by refining one of the bins into ten finer bins of width $0.1^{i+1}$, and so on until the last digit is processed (see Figure 3). The top right of the figure shows an example of a time series serialized as an input string.

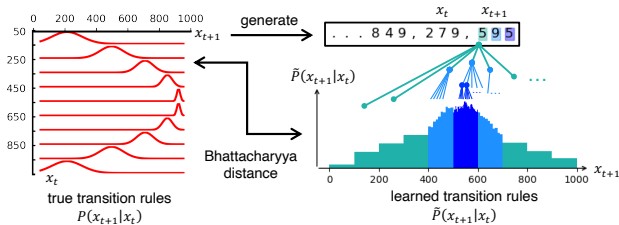

*Figure 3.* An example of hierarchical transition rules extracted from LLaMa-13b. The PDF bins are color-coded based on resolutions, which in this example are more refined near the mode. The height of $\tilde{P}(x_{t+1}|x_t)$ is shown in log scale.

**Hierarchy-PDF algorithm.** While a single pass through the LLM yields a discretized PDF represented by bins of various widths, we can refine the PDF by querying each coarse bin. For example, to furnish a maximal resolution PDF of a 3-digit number, we need to query all $10^2$ combinations of the first two digits of that number. Suppose a time series consists of $S$ values (steps), each represented as $n$ digits. Obtaining a maximal resolution PDF for each value of the entire sequence requires $10^{n-1}S$ forward passes of the LLM. This daunting process could be significantly simplified because most of the $10^{n-1}S$ inputs differ only in the last tokens, and thus one can recursively cache the key and value matrices associated with the shared tokens. The computation can be further reduced by refining only the high-probability bins near the mode, which dominate the loss functions, as shown in Figure 3. Algorithm 1 outlines the *Hierarchy-PDF* algorithm used to recursively refine the PDF associated with a multi-digit value in a time series (more details available in Appendix A).

## 4. Experiments and Results

This section reports empirical in-context learning results on two example systems: discrete Markov chain and stochastic logistic map. We defer discussion of other systems, reported in Figure 1, to Appendices A.3 and A.4. The experiments are repeated ten times with trajectories initiated by different random seeds.

**Model choice.** All numerical experiments reported in this section are performed using the open-source LLaMA-13b model. While we observe that larger language models, such

---

**Algorithm 1** Hierarchy-PDF

**Input:** Unrefined PDF, current depth $D_c$, target depth $D_t$
**Procedure:** RecursiveRefiner(PDF, $D_c$, $D_t$)
  **if** $D_c = D_t$ **then**
    end the recursion
  **else if** current branch is refined **then**
    Alter the last digit to launch 9 recursive branches
    RecursiveRefiner(PDF, $D_c$, $D_t$)
  **else if** current branch is unrefined **then**
    refine PDF with new logits
    **if** $D_c + 1 < D_t$ **then**
      Append the last digit to launch 10 recursive branches
      RecursiveRefiner(PDF, $D_c + 1$, $D_t$)
    **end if**
  **end if**
**Output:** Refined PDF

---

as LLaMA-70b, may achieve lower in-context loss on some dynamical systems (Appendix A.3.3), they do not display qualitative differences and affect our conclusions.

### 4.1. Markov chains with discrete states

The transition rules of a time-independent Markov chain with $n$ states consist of a stochastic matrix $(P_{ij})_{1 \le i,j \le n}$, defined as

$$P_{ij} = P(X_{t+1} = j | X_t = i), \quad 1 \le i, j \le n.$$

Using the testing procedures elaborated in Section 3.2, we generate 10 Markov chains, each from a distinct and randomly generated transition matrix of size $n = 4$.

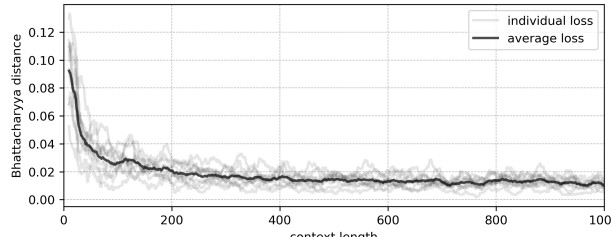

*Figure 4.* Markov chain in-context loss curves decay rapidly with respect to the input time series length. The average loss is obtained from 10 individual loss curves.

The corresponding loss curves between the LLM predictions and the ground truth are displayed in Figure 4. The LLM formulates remarkably accurate statistical predictions as more time steps are observed in context, even though the transition rules are synthesized completely at random. These conclusions hold for larger transition matrices ($n > 4$) and more sophisticated LLMs, such as LLaMA-70b (see Appendix A.3.3).

## 4.2. Noisy logistic map

The logistic map, first proposed as a discrete-time model for population growth, is one of the simplest dynamical systems that manifest chaotic behaviors (Strogatz, 2015). It is governed by the following iterative equation:

$$x_{t+1} = f(x_t) = rx_t(1 - x_t), \quad x_0 \in (0, 1),$$

where $r \in [1, 4)$ is a parameter. The logistic map system becomes stochastic when one introduces small Gaussian perturbations of variance at each step, resulting in modified iterative equation:

$$x_{t+1} = f(x_t + \epsilon),$$

where $\epsilon \overset{i.i.d}{\sim} \mathcal{N}(0, \sigma^2)$. In this case, the ground truth distribution of the next state, $x_{t+1}$, conditioned on the current state $x_t$ is Gaussian with mean $f(x_t)$ and variance $(\sigma f'(x_t))^2$:

$$X_{t+1}|\{X_t = x_t\} \sim \mathcal{N}\left(f(x_t), (\sigma f'(x_t))^2\right). \quad (1)$$

The first derivative of $f$ measures how sensitive the local dynamics are to external perturbations. This intuitively explains why the standard deviation of the conditional distribution should be proportional to $f'$. We note that the approximation in Equation (1) assumes a small perturbation compared to the second derivative, that is $\sigma^2 \ll 1/f''(x)$.

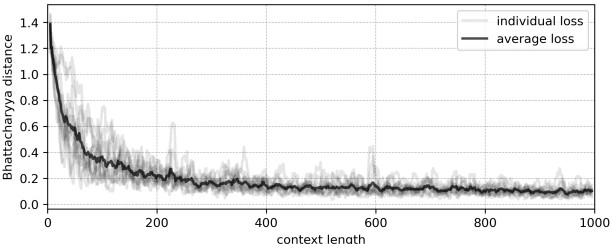

*Figure 5.* Stochastic logistic map in-context loss curves.

We again observe a power-law-like decay of the in-context loss function with respect to the length of the observed time series in Figure 5. To achieve low in-context loss, the LLM must learn to predict not only the mean, but also the variance of future steps. This is shown in Figure 6 and discussed further in Appendix A.9.

## 5. Discussion and conclusion

**Data leakage.** The possibility that LLMs' accurate predictions of the next time step value are merely due to memorization seems extremely unlikely. A sequence of even one thousand numerical values, encoded with three digits, may cover $10^{3000}$ particular instances, which is well beyond the $\sim 10^{12}$ tokens of the training corpus (Touvron et al., 2023).

**In-context neural scaling law.** Neural scaling laws (Kaplan et al., 2020) are power laws that characterize how the loss

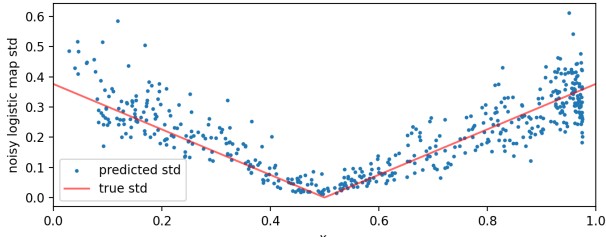

*Figure 6.* Noisy logistic map standard deviation as a function of the state value $x_t$, learned by the LLM, along with the ground truth.

of trained neural networks vary with respect to parameters of the model, such as model size, dataset size, and computational resources. To the best of our knowledge, neural scaling laws have so far only been observed in the training procedure, which updates the weights of neural networks using an explicit algorithm, such as stochastic gradient descent and Adam (Kingma & Ba, 2014). The loss curves observed in the different numerical experiments (see Figures 1 and 18) reveal an additional *in-context* scaling law for LLMs zero-shot learning of dynamical systems. Further analysis of these scaling laws are resented in Appendix A.6.

**Main conclusions.** We showed that, with increasing cardinality in tokenization, and given sufficient context, LLMs can accurately extrapolate not just deterministic trajectories (Appendix A.4), but also chaotic and stochastic sequences governed by specific transition rules. In the latter case, the extrapolation accuracy is measured in a statistical sense. This suggests that large language models can in-context learn dynamical systems' time series and predict future states in a manner that maintains fidelity with the underlying principles governing the system's evolution. Moreover, this behavior is observed without the use of any fine-tuning or "prompt engineering". Our results suggest that a transformer, trained primarily on textual data, can extract governing principles of numerical sequences observed in-context.

**Future directions.** The in-context neural scaling law hints at a learning algorithm that LLMs implicitly implement during inference, such as gradient descent (Von Oswald et al., 2023). Characterizing such an algorithm is an open question of broad interest (Shen et al., 2023). Another exciting future direction is to study the generalization of the observed in-context neural scaling laws for other LLMs, such as GPT4 (OpenAI, 2023), and the newly proposed state space models (Gu & Dao, 2023).

## Software and Data

The code and data supporting this study are available at: https://github.com/AntonioLiu97/llmICL.

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

# A. Appendix

## A.1. Loss Functions

Once the learned transition rules, $\tilde{P}(X_{t+1}|X_t)$, have been extracted, we quantify the deviation from the ground truth $P(X_{t+1}|X_t)$. Depending on the nature of the system, one of the following two loss functions may be more appropriate (see Section 4).

**Bhattacharyya distance.** For stochastic time series, we use the Bhattacharyya distance to characterize the distance between learned and ground truth transition functions. The Bhattacharyya distance between $P$ and $\tilde{P}$, on a domain $\mathcal{X}$ is defined as (Bhattacharyya, 1943; 1946; Kailath, 1967):

$$D_{B}(P, \tilde{P}) = -\ln\left(\int_{\mathcal{X}} \sqrt{p(x)\tilde{p}(x)}\mathrm{d}x\right), \tag{2}$$

and has been widely employed by feature selection and signal extraction methods (Choi & Lee, 2003; Kailath, 1967). Since $\tilde{P}(X_{t+1}|X_t)$ takes the form of a hierarchical PDF, one may approximate the integral in Equation (2) via a discrete quadrature rule as

$$D_{B}(P, \tilde{P}) = -\ln\left(\sum_{x} \sqrt{p(x)\tilde{p}(x)}\Delta x\right), \tag{3}$$

where $\Delta x$ denotes the length of the sub-interval containing $x$ in the partition of $\mathcal{X}$.

**Squared deviations from the mean.** For deterministic systems, the true transition functions become delta-functions. As a result, the discretized Bhattacharyya distance from Equation (3) reduces to (see Equation (5) in Appendix A.2)

$$D_{B}(\delta(x - x_{\text{true}}), \tilde{P}) = -\frac{1}{2}\ln(\tilde{p}(x_{\text{true}})) + C,$$

which is proportional to the negative log-likelihood (NLL) assigned to the true data by the LLM, plus a constant $C$[2]. NLL references only the finest bins in the hierarchical PDF and is thus unstable as in-context loss. As an alternative, we use the squared deviations from the mean (SDM) (Kobayashi & Salam, 2000) as the in-context loss for deterministic systems:

$$\text{SDM}(x_{\text{true}}, \tilde{P}) = \left(x_{\text{true}} - \sum_{x \in \mathcal{X}} \tilde{p}(x)x\Delta x\right)^2,$$

where the mean $\mu_{\tilde{P}} = \sum_{x} \tilde{p}(x)x\Delta x$ is extracted from the hierarchical PDF. Note that unlike the Bhattacharyya distance, which references the model prediction $\tilde{p}$ only at $x_{\text{true}}$, the SDM takes into account the entire support $x \in \mathcal{X}$. Our numerical experiments suggest that SDM is more stable and better captures the in-context learning dynamics of deterministic systems (see Appendix A.4).

## A.2. Additional loss functions

**KL-divergence.** The KL-divergence between two PDFs, $P$ and $\tilde{P}$, is defined as

$$D_{KL}(P, \tilde{P}) = \sum_{x \in \mathcal{X}} P(x) \log\left(\frac{P(x)}{\tilde{P}(x)}\right). \tag{4}$$

Although commonly used as the training loss for a variety of machine learning systems, this loss function may suffer from numerical instabilities as the learned transition function $\tilde{P}$ are often close to zero, as shown in Figures 8 and 13, where the probability density is concentrated in small regions of the support.

**Discretized Bhattacharyya distance for deterministic systems.** For deterministic systems, the ground truth transition function is a delta function. Therefore, the Bhattacharyya distance between it and the Hierarchy-PDF prediction only references the finest bin associated with the true value $x_{\text{true}}$.

$$D_{B}(\delta(x - x_{\text{true}}), \tilde{P}) = -\ln\left(\sum_{x} \sqrt{\delta(x - x_{\text{true}})\tilde{p}(x)}\Delta x\right) = -\frac{1}{2}\ln(\tilde{p}(x_{\text{true}})) - \ln \Delta x$$

$$= -\frac{1}{2}\ln(\tilde{p}(x_{\text{true}})) + \text{constant}. \tag{5}$$

---

[2]This constant is determined by the base B of the system, and the number of digits n as $C = -\ln \Delta x = n \log B$.

As a result, the Bhattacharyya distance is reduced to an affine-transformed negative log-likelihood assigned to data by the LLM. Such local sensitivity on $\tilde{p}(x_{\text{true}})$ explains the wild fluctuations seen in the Bhattacharyya loss in Appendix A.4.

**Higher moments and kurtosis.** While the Bhattacharyya distance and SDM measure the agreement between the extracted transition rules $\tilde{P}$ and the ground truth distribution $P$, they do not explicitly characterize the type of the distribution (e.g., Gaussian or uniform). We employ the kurtosis as an additional measure to assess whether the LLM recovers the correct shape of $P$. The kurtosis of a distribution $P$ is defined as (Joanes & Gill, 1998)

$$\text{Kurt}(P) = \frac{\mathbb{E}_{x \sim P}[(x - \mu_P)^4]}{\mathbb{E}_{x \sim P}[(x - \mu_P)^2]^2} = \frac{\mu_4}{(\sigma^2)^2}, \tag{6}$$

where $\sigma^2$ and $\mu_4$ are the second and fourth central moments, which can be approximated using a hierarchical PDF as

$$\sigma^2(P) = \sum_x p(x)(x - \mu_p)^2 \Delta x, \quad \mu_4(P) = \sum_x p(x)(x - \mu_p)^4 \Delta x. \tag{7}$$

The kurtosis is equal to 3 for a Gaussian distribution and $9/5$ for bounded uniform distributions. Figure 7 shows the kurtosis of Brownian motion transition rules learned by LLM, which converges to 3 as the context length increases.

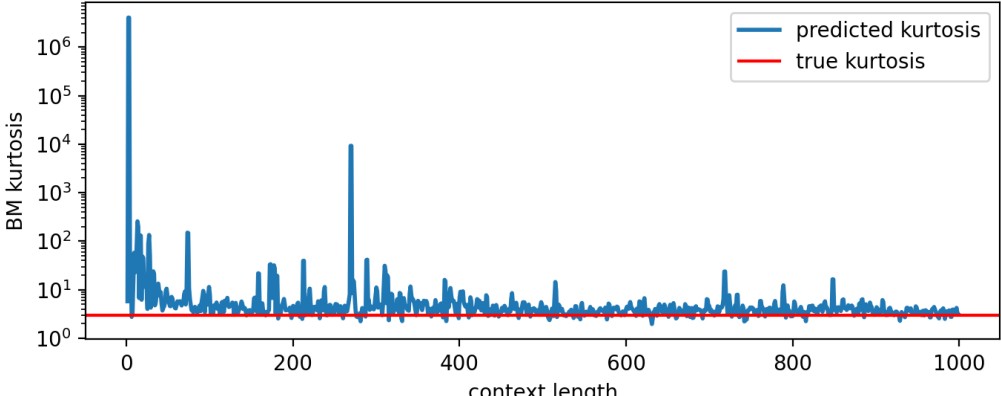

*Figure 7.* Kurtosis of Brownian motion transition rules with respect to the input length. Blue: kurtosis of LLM predicted PDF. Red: ground truth kurtosis, which is 3 for all Gaussian distributions.

### A.3. Additional Experiments: stochastic time series

#### A.3.1. BROWNIAN MOTION

Brownian motion is an example of a continuous-time stochastic process (Einstein, 1905), and is described by a stochastic differential equation (SDE):

$$dX_t = \mu dt + \sigma dW_t, \tag{8}$$

where $X_t$ represents the state of the system at time $t$, $\mu$ is the drift coefficient, $\sigma$ is the volatility coefficient, and $dW_t$ is the increments of a Wiener process (Revuz & Yor, 2013), modeling the randomness of motion.

To simulate trajectories of Brownian motion, we use the Euler–Maruyama method (Platen, 1999), which discretizes Equation (8) as $X_{t+\Delta t} = X_t + \mu \Delta t + \sigma \sqrt{\Delta t} Z$, where $\Delta t$ is the time resolution, and $Z \sim \mathcal{N}(0, 1)$ is a random variable that follows a standard Gaussian distribution. The Euler–Maruyama method may also be written as a conditional distribution:

$$X_{t+\Delta t} | \{X_t = x_t\} \sim \mathcal{N}(x_t + \mu \Delta t, \sigma^2 \Delta t),$$

which is the ground truth transition function visualized in Figure 8. Indeed, the ground truth next state is described as a Gaussian distribution, and we observe in Figure 8 that the LLM prediction agrees well with the true, underlying distribution. Additionally, as shown in Figure 8, the LLM displays the correct Gaussian shape for the PDF, converging to a measured kurtosis of 3 (see Appendix A.2). We then simulate ten different trajectories using random seeds for $Z$ and report the resulting LLM learning curves in Figure 9, measured in the Bhattacharyya distance.

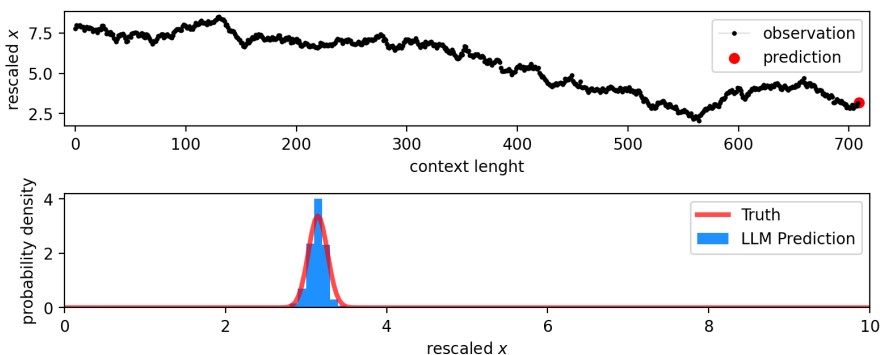

*Figure 8.* Next state prediction of Brownian motion. Top: Input stochastic time series shown in black, and the state to be predicted is highlighted in red. Bottom: The LLM's prediction, along with the ground truth distribution.

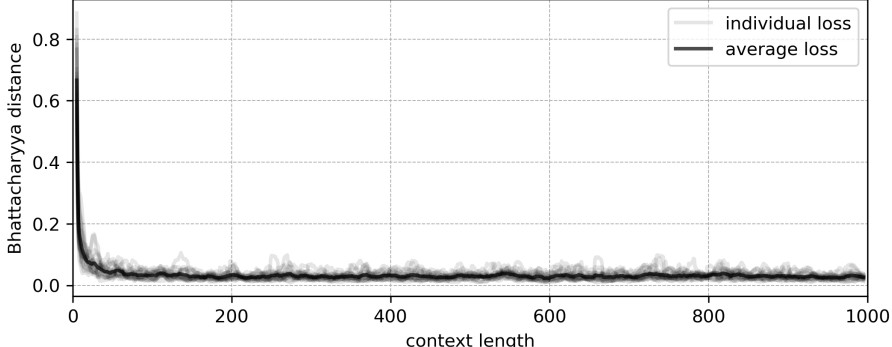

*Figure 9.* Bhattacharyya distance between the LLM predicted PDF and the ground truth transition function of Brownian motion with respect to the input length.

### A.3.2. GEOMETRIC BROWNIAN MOTION

Geometric Brownian motion (GBM) (Oksendal, 2013) is a stochastic process that is commonly used in mathematical finance to model the trajectories of stock prices and other financial assets (Hull, 2021). A GBM is governed by the following SDE:

$$dX_t = \mu X_t dt + \sigma X_t dW_t, \tag{9}$$

where $X_t$ models the price of an asset at time $t$, and the fluctuation term $\sigma X_t dW_t$ is proportional to the current asset price $X_t$. The Euler–Maruyama discretization of the GBM reads $X_{t+\Delta t} = X_t + \mu X_t \Delta t + \sigma X_t \sqrt{\Delta t} Z$, and leads to the ground truth transition function:

$$X_{t+\Delta t}|\{X_t = x_t\} \sim \mathcal{N}(x_t + \mu x_t \Delta t, (\sigma x_t)^2 \Delta t). \tag{10}$$

We simulate ten different GBM trajectories using random seeds and report the corresponding learning curves in Figure 10.

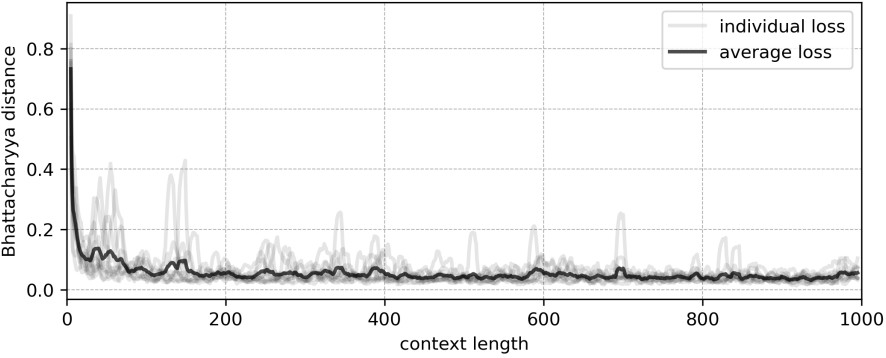

*Figure 10.* Geometric Brownian motion in-context loss curve.

We perform an additional numerical test to verify that the LLM is learning the correct relationship between the variance of the GBM and the state value $X_t$ (see Equation (10)). To investigate this, we display in Figure 11 the expected standard deviation along with the learned one, extracted from the Hierarchy-PDF using Equation (7), across all predicted states. We find that the LLM respects the ground truth standard deviation of the GBM, as prescribed by the underlying transition function.

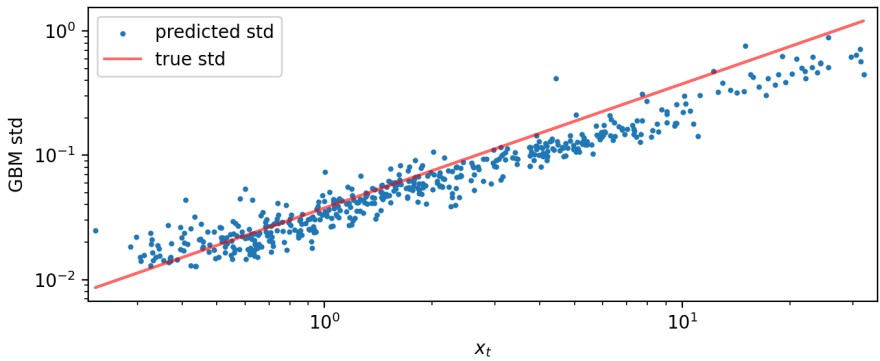

*Figure 11.* Evolution of the geometric Brownian motion standard deviation with respect to the state value $x_t$ (see Equation (10)), along with the predicted standard deviation extracted from the LLM at each time step.

### A.3.3. MARKOV CHAINS WITH LLAMA-70B

Our experiments show that LLMs generally achieve lower in-context loss for Markov chains with fewer discrete states $n$, as shown in Figure 12. For both LLaMA-13b and LLaMA-70b, the in-context loss curves cease to decrease significantly for numbers of states $n \geq 9$.

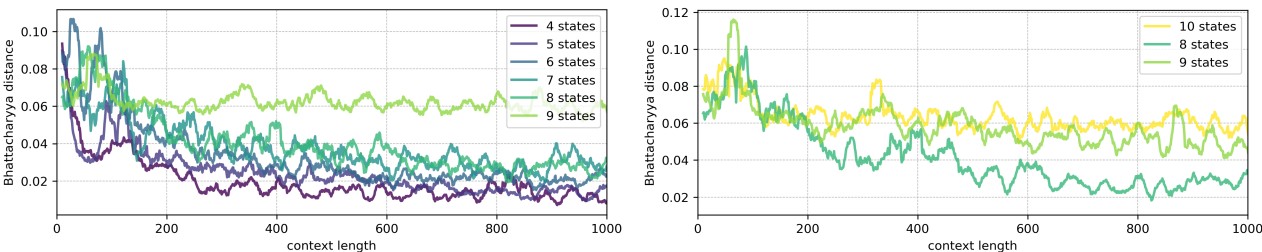

*Figure 12.* In-context loss curves for LLaMA-13b (left) and LLaMA-70b (right) with respect to the number of states in the transition matrix.

### A.4. Additional Experiments: deterministic time series

#### A.4.1. LOGISTIC MAP

The logistic map, first proposed as a discrete-time model for population growth, is one of the simplest dynamical systems that manifest chaotic behavior (Strogatz, 2015). It is governed by the following iterative equation:

$$x_{t+1} = f(x_t) = rx_t(1 - x_t), \quad x_0 \in (0, 1), \tag{11}$$

which may also be written using conditional distributions to reflect the deterministic nature of the system as $X_{t+1}|\{X_t = x_t\} \sim \delta_{f(x_t)}$, where $\delta$ denotes the Dirac delta distribution. This conditional distribution is the ground truth transition function displayed in red in Figure 13. The parameter $r \in [1, 4)$ controls the behavior of the system and is set to $r = 3.9$. At this value, the dynamics are naturally confined within the interval $(0, 1)$, and the system has no stable fixed points. Due to the chaotic nature of the system, two initial nearby trajectories diverge exponentially in time. This property allows us to sample multiple uncorrelated trajectories by using different initial conditions $x_0$, sampled uniformly in $(0, 1)$.

Figure 13 displays one of the ten tested trajectories and an LLM's prediction of the last state. The PDF of the next state prediction is extracted using the Hierarchy-PDF algorithm described in Section 3.2. We find that the LLM prediction is

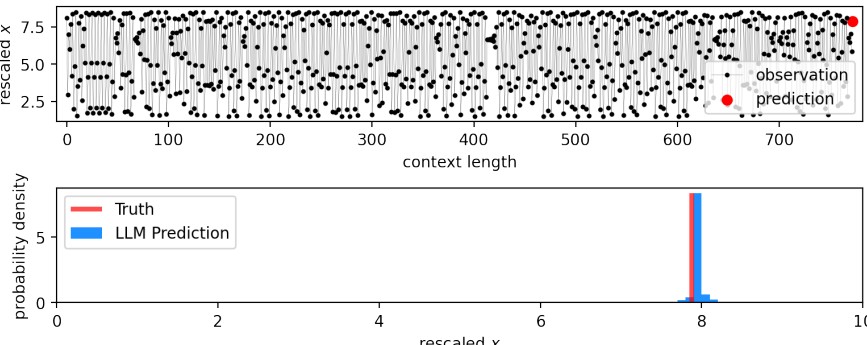

*Figure 13.* Next state prediction of the logistic map. Top: Input chaotic time series shown in black, and the state to be predicted is highlighted in red. Bottom: The LLM's statistical prediction for the last state. The ground truth distribution is delta-distributed, which is shown as a vertical red line.

close to the ground truth, except for minor deviations manifested by small, but non-zero, probability densities in neighboring values. While the extracted prediction is only reported for the last time step in the bottom panel of Figure 13, we also extract the model prediction at every time step for all tested trajectories and report the corresponding Bhattacharyya and SDM losses in Figure 14.

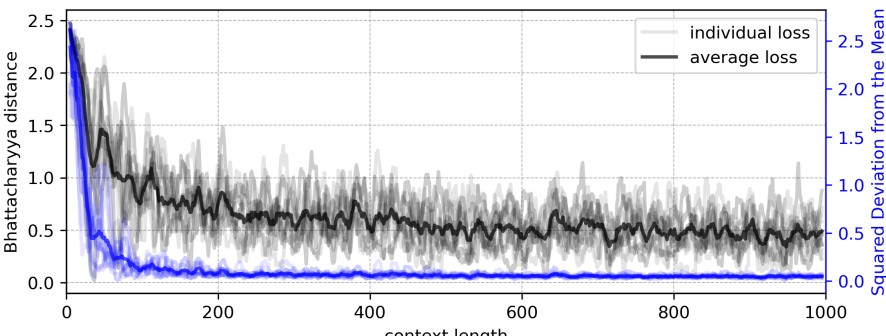

*Figure 14.* Logistic map in-context loss curves. For deterministic systems, Bhattacharyya loss is subject to large fluctuations while SDM loss is more stable.

As foreshadowed in Section 3, the Bhattacharyya loss suffers from large fluctuations with deterministic systems such as the logistic map, while the SDM loss better captures the in-context learning dynamics. In particular, the SDM loss decreases rapidly with the number of observed states, without any fine-tuning nor prompt engineering of the LLM. This suggests that the LLM can extract the underlying transition rules of the logistic map from in-context data.

### A.4.2. LORENZ SYSTEM

The Lorenz system (Lorenz, 1963) is a three-dimensional (3D) dynamical system derived from a simplified model of convection rolls in the atmosphere. It consists of a system of three ordinary differential equations:

$$\dot{x}(t) = \sigma(y - x), \quad \dot{y}(t) = x(\rho - z) - y, \quad \dot{z}(t) = xy - \beta z,$$

where $\sigma = 10$, $\rho = 28$ and $\beta = 8/3$ are parameters dictating the chaotic behavior of the system. We compute ten 3D trajectories using a first-order explicit time-stepping scheme. All trajectories share the same initial conditions in $y$ and $z$, and differ only in the $x$-coordinate, which is uniformly sampled in $(0, 0.3)$. The chaotic nature of the system guarantees that the sampled trajectories quickly diverge from one another. We prompt the LLM with the $x$-component of the simulated series and extract the next predicting values.

When the $x$, $y$, and $z$ components are observed, the system is deterministic and Markovian; in the sense that a state vector $\vec{s}_t = (x_t, y_t, z_t)$ at time $t$ fully determines the next state $\vec{s}_{t+1}$. However, if the $x$-component is the only one observed, then the system ceases to be Markovian but remains deterministic if one expands the state vector to include information from

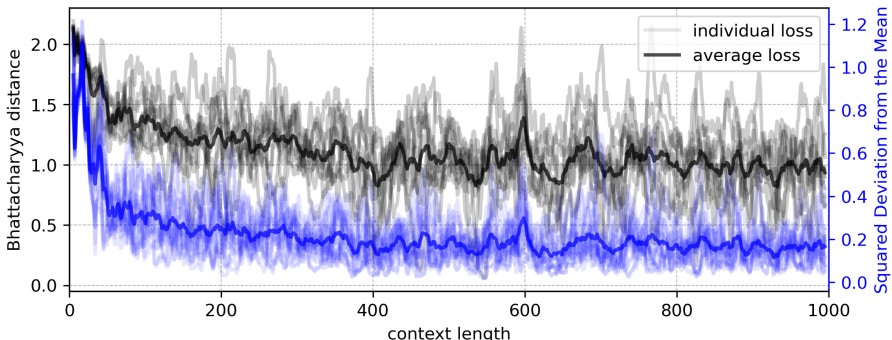

*Figure 15.* Loss curves for predicting the $x$-component of the Lorenz system with respect to the number of observed time steps.

earlier states. Hence, Takens' embedding theorem (Takens, 2006) guarantees that the observation of at most seven states of the series $x_{0:t}$ is sufficient to predict $x_{t+1}$. Finding the optimal number of states to reconstruct the system's trajectory is an area of active research (Strogatz, 2015). Despite this apparent difficulty, LLaMA-13b can formulate increasingly accurate predictions of the series as it observes more states, as evidenced by the decaying loss curves plotted in Figure 15.

### A.5. Continuous State Space Visualization

One may naively remark upon the possibility that the in-context learning task for the Lorenz system and the logistic map could be rendered trivial if $x_{t+1}$ always falls close to $x_t$, in which case the LLM only needs to learn a static noisy distribution in order to decrease the loss. This is not the case with our experiments. In this section, we demonstrate the non-triviality of the learning tasks in Figures 16 and 17. In both cases, it is clear that the LLM has successfully learned to actively predict the expected mean position of the next state, and, in the logistic map example, the variance of the next state distribution as well. We note that the Lorenz system is simulated deterministically, hence the true next-state distribution is represented as a delta-function.

### A.6. In-context neural scaling law

Neural scaling laws (Kaplan et al., 2020) describe how the performance of trained neural networks, particularly language models, scales with changes in key factors such as model size ($N$), dataset size ($D$), and computational resources used for training (C). These laws are often observed as power-law relationships in the following form:

$$L(N) = \left(\frac{N}{N_c}\right)^{\alpha_N}, \quad L(D) = \left(\frac{D}{D_c}\right)^{\alpha_D}, \quad L(C) = \left(\frac{D}{C_c}\right)^{\alpha_C},$$

where $L$ represents the loss or performance metric of the model. The characteristic factors ($N_c$, $D_c$, and $C_c$), and power coefficient ($\alpha$) are extracted empirically from training curves. The fitted quantities depend on the distribution of data, the model architecture, and the type of optimizer used for training. Such power-law relations appear in log-log plots as straight lines, whose slopes correspond to the parameter $\alpha$. Our loss curves from learning dynamical systems (see Figure 1) reveal an additional neural scaling law that applies to in-context learning:

$$L(D_{in}) = \left(\frac{D_{in}}{D_c}\right)^{\alpha},$$

where $D_{in}$ stands for the length of time series observed in the prompt (in-context). In Figure 18, we display the fitted power laws to the in-context loss curves.

### A.7. Baselines for noisy logistic map and Markov chains

In this section, we compare LLM's predictions against baseline models of known architectures, in order to understand the difficulty of the in-context learning task and make better sense of the Bhattacharyya loss. Specifically, we consider the following baseline models: unigram and bi-gram models for discrete Markov chains, and linear and non-linear autoregressive models with 1-step memory (AR1) for noisy logistic maps. The bi-gram model for the Markov chain has an unfair advantage since it is designed to model Markovian processes where the probability distribution of a token depends only on the previous

*Figure 16.* 4 consecutive states in a noisy logistic map. Ground truth is shown in red and LLaMA predictions in blue.

*Figure 17.* 4 consecutive states in a Lorenz system trajectory. Ground truth is shown in red and LLaMA predictions in blue.

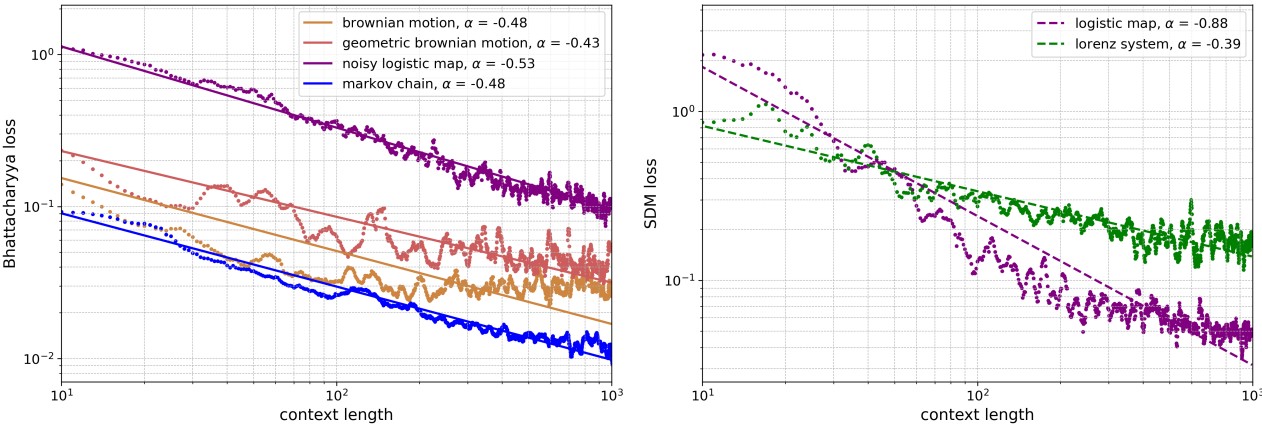

*Figure 18.* In-context loss curves from LLaMA-13b fitted with power law, with fitted power coefficient $\alpha$ shown in legend. Left: loss of stochastic series measured in Bhattacharyya distance. Right: loss of deterministic series measured in SDM.

token, ie., inferring the values of the transition matrix. The unigram model, on the other hand, models all tokens as drawn i.i.d. from the same distribution.

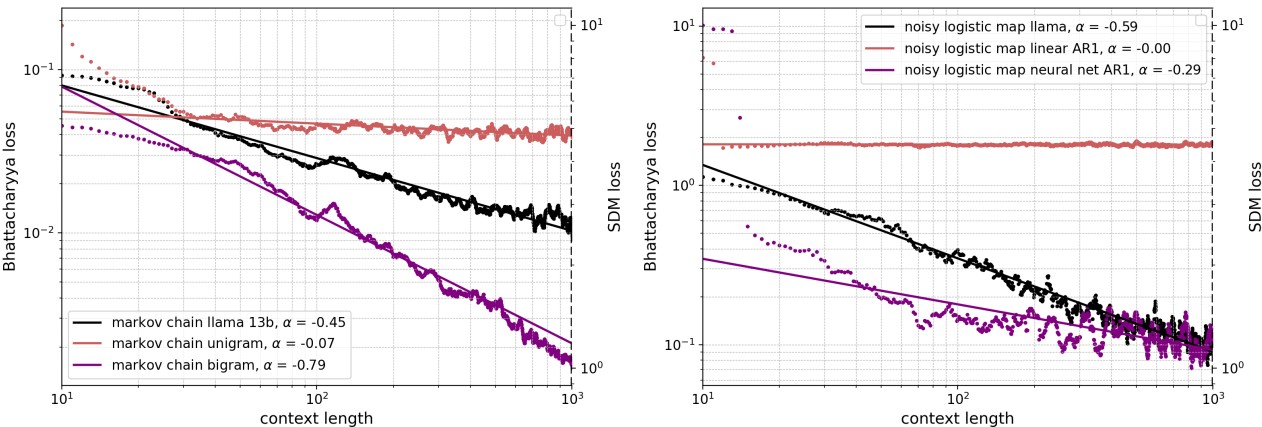

*Figure 19.* LLM in-context loss curves against the baseline model loss curves. The coefficient $\alpha$ denotes the fitted scaling exponent.

The neural network AR1 model takes a state $x_{t-1}$ as input, and outputs prediction for next state $x_t$ as a Gaussian distribution parameterized by mean and variance: $f_\theta : x_t \to \mathcal{N}(\mu_\theta(x_t), \sigma_\theta(x_t))$. As such, it also has the unfair advantage of hard-coded Gaussianiety. LLaMA, on the other hand, must infer the correct distribution family from data. Despite this intrinsic disadvantage, LLaMA still outperforms the neural network AR1 model in the large context limit. The NN used in the non-linear AR1 models features three fully connected hidden layers of widths 64, 32, and 16. We found that simpler neural networks are easily trapped in local minima, leading to unstable performance. The loss curves in Figure 19 are obtained by training an independent copy of this neural network to convergence at each context length, and predict the next state distribution using the trained NN. The training loss is defined as the negative log-likelihood of the observation data:

$$\mathcal{L}(\text{data}, \theta) = - \sum_{x_{t-1}, x_t \in \text{data}} \log P(x_t; \mu_\theta(x_{t-1}), \sigma_\theta(x_{t-1}))$$

$$= \sum_{x_{t-1}, x_t \in \text{data}} \log \sigma_\theta(x_{t-1}) - \frac{1}{2} \left( \frac{x_t - \mu_\theta(x_{t-1})}{\sigma_\theta(x_{t-1})} \right)^2 .$$

Furthermore, the ensemble of NNs allows us to visualize the learned transition functions $P(x_{t+1}|x_t)$ at each context length. In Figure 20, we show how the transition rules learned by the NNs gradually converge to the ground truth as context length increases. Since at large context length, the LLMs achieve similar loss as the NN-based AR1 model, it is reasonable to expect the LLM to have learned a transition function of similar accuracy as shown in the 5th plot in Figure 20. However, it is difficult to visualize the full transition rules $P(x_{t+1}|x_t)$, for $x_t \in [0, 1]$, as learned by an LLM, because doing so would require appending an array of $x_t$s at the end of a training sequence, which would render the training sequence incorrect.

### A.8. Invariant measure and the early plateauing of in-context loss

While most datasets are well-described by the power laws, two loss curves — the Brownian motion and geometric Brownian motion — plateau early at a context length of about $10^2$, as shown in Figure 18. We attribute this early plateauing to the fact that the Brownian and geometric Brownian motions "wander out of distribution" at large time $t$, while all other dynamical systems studied in this paper converge to stable distributions (i.e., the invariant measure). A Markovian system (stochastic or deterministic) governed by a transition rule $P(x_{t+1}|x_t)$ is said to have an invariant measure $\pi$ if

$$\pi(x_{t+1}) = \int_{\mathcal{X}} \pi(x_t) P(x_{t+1}|x_t) \, dx_t, \quad x_{t+1} \in \mathcal{X}. \tag{12}$$

If a system is initialized by $\pi(x)$ and evolves according to $P$, then the distribution of states at the next step will still follow $\pi(x)$. This property makes $\pi$ an invariant or stationary distribution for the system. It has been shown that the logistic map

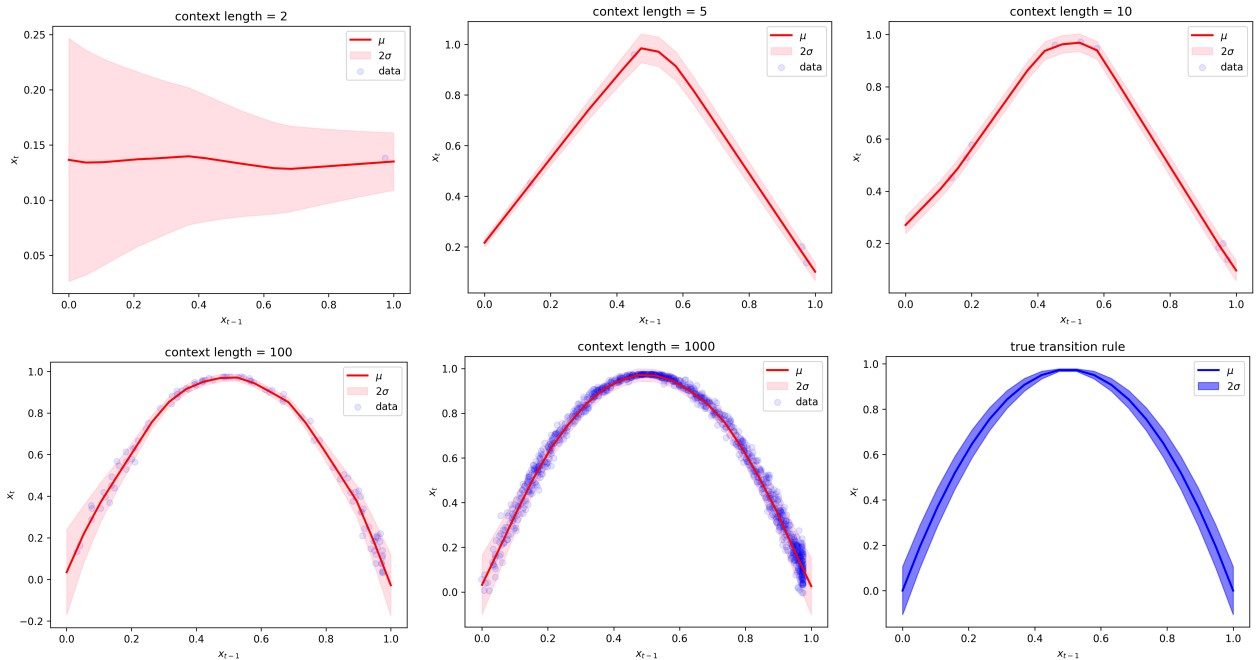

*Figure 20.* Noisy logistic map transition rules, $P(x_t | x_t - 1)$, learned by a neural network-based AR1 model against the ground truth transition rule.

and Lorenz systems in the chaotic regime converge almost surely to their respective invariant measure, regardless of the initialization (Strogatz, 2015).

For discrete Markov chains governed by a transition matrix $p$, the stationary distribution is defined as a discrete probability mass function, denoted by $\vec{\pi}$, such that

$$\vec{\pi} = p\vec{\pi}, \tag{13}$$

which is analogous to the continuous case described by Equation (12). By definition, any non-negative right eigenvector of $p$ with eigenvalue $\lambda = 1$ is a stationary distribution of $p$. (Sethna, 2021) showed that a valid transition matrix has at least one stationary distribution. On the other hand, neither the Brownian nor the geometric Brownian motion has invariant distributions[3] on unbounded domains (e.g., when $\mathcal{X} = \mathbb{R}$). This can be seen from the marginalized distribution $P(x_t)$ at time $t$. For the Brownian motion defined in Equation (8), the marginalized distribution of $x_t$ at time $t$ is a normal distribution:

$$P(x_t) = \frac{1}{\sqrt{2\pi\sigma^2 t}} \exp\left(-\frac{(x_t - \mu t)^2}{2\sigma^2 t}\right), \tag{14}$$

while for the geometric Brownian motion defined in Equation (9), the marginalized distribution of $x_t$ is a log-normal distribution (Crow & Shimizu, 1987):

$$P(x_t) = \frac{1}{x_t\sqrt{2\pi\sigma^2 t}} \exp\left(-\frac{(\log(\frac{x_t}{x_0}) - \mu t - \frac{\sigma^2}{2}t)^2}{2\sigma^2 t}\right). \tag{15}$$

Both Equations (14) and (15) are time-dependent and do not converge to a stationary distribution in the limit $t \to \infty$. For the Brownian and geometric Brownian motions, the LLM might decide to only consider the most recent segment of time steps, and ignore the earlier data, which are in some sense "out of distribution". This could explain the early plateauing of loss curves. Indeed, the classical neural scaling laws can be improved or broken if the scheduling of the training data shifts in distribution, as shown in (Sorscher et al., 2023). Different from (Sorscher et al., 2023; Lu et al., 2022), which alter the scheduling of data to achieve better learning curves that decrease faster with the size of training data, our experiments consider time series with pre-determined transition laws. We cannot tamper with the scheduling of our data to make it stationary without altering the underlying transition rules.

---

[3]For stochastic systems, the invariant measure is sometimes referred to as the stationary distribution.

## A.9. Temperature and variance

The temperature $T$ is a hyper-parameter that controls the variance of the softmax output layer. Although most LLMs are trained at $T = 1$, it is common practice to tune the temperature in the interval $T \in [0.8, 1.2]$ during inference. Then, one can opt for increased diversity (high $T$), or better coherence (low $T$) in the generated output. The temperature hyper-parameter affects the uncertainty, or variance, in the Hierarchy-PDF extracted from the LLM. Figures 21 and 22 show how different temperatures change the shape of the Hierarchy-PDF. In both cases, higher temperature leads to higher variance in the PDF.

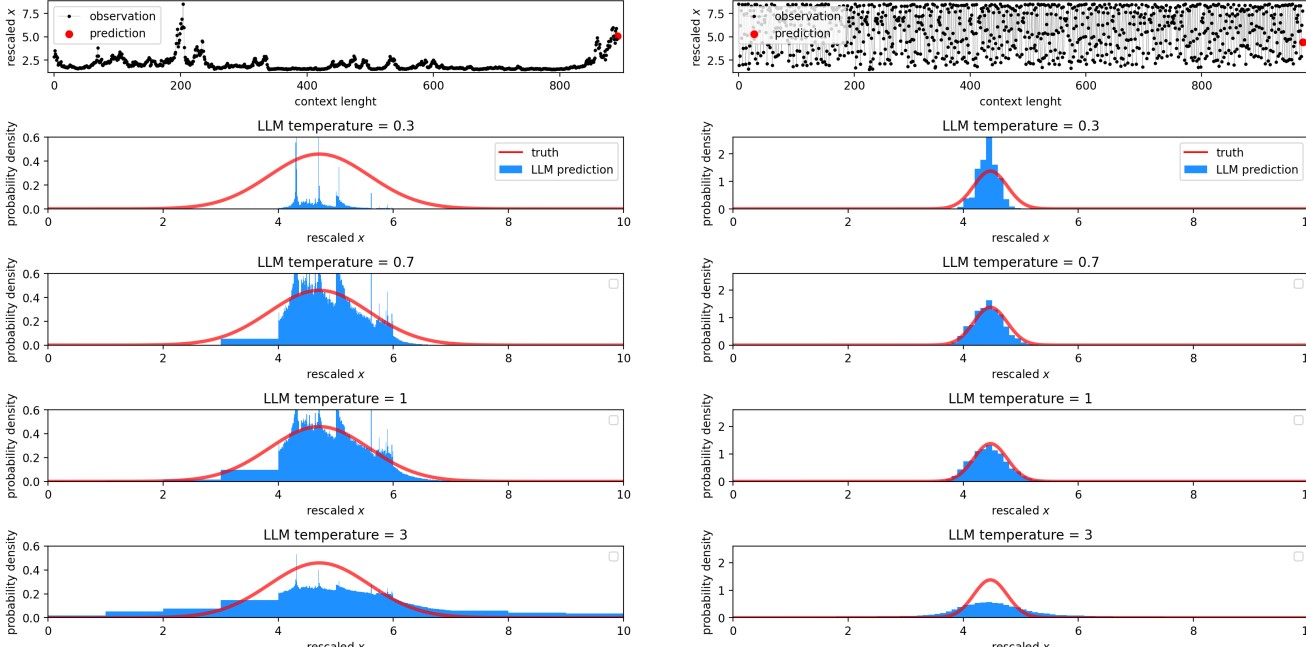

*Figure 21.* Next state prediction for Geometric Brownian motion. Topmost: Input stochastic time series (black), and the state to be predicted (red). Rest: The Hierarchy-PDF prediction extracted from LLaMA-13b evaluated at different temperatures ranging from $T = 0.3$ to 3, along with ground truth PDF (red).

*Figure 22.* Next state prediction for the noisy logistic map. Topmost: Input stochastic time series (black), and the state to be predicted (red). Rest: The Hierarchy-PDF prediction extracted from LLaMA-13b evaluated at different temperatures ranging from $T = 0.3$ to 3, along with ground truth PDF (red).

We highlight the different refinement schemes used in these figures: for GBM, the PDF is refined to the last (third) digit near the mode, and left coarse elsewhere. This is because the true variance for GBM can span two orders of magnitude (see Figure 11), with most data points trapped in the low-variance region at small $X_t$. Hence, we require high precision to resolve these small variances in Figure 23. On the other hand, the noisy logistic map time series does not suffer from this issue, and thus we uniformly refine its PDF only up to the second digit.

While the loss curves in our paper are calculated at $T = 1$, the predicted $\sigma$ shown in Figures 6 and 11 are extracted at $T = 0.7$. We performed a grid search on the temperature ranging from $T = 0.3$ to $T = 3$ (see Figures 24 and 25), and observed that $T = 0.7$ consistently results in better prediction quality of the variance.

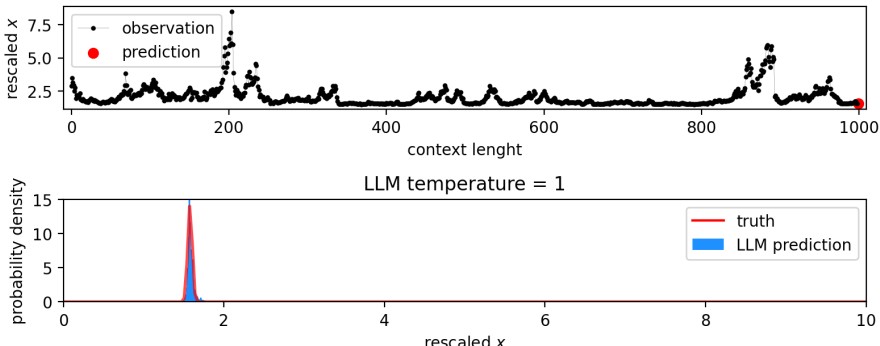

*Figure 23.* Most data points in GBM are trapped in low variance region with small $X_t$. The hierarchy-PDF must be very refined to resolve these minuscule variances.

*Figure 24.* GBM standard deviation $\sigma$ as a function of state value $X_t$, learned by the LLM, along with the ground truth. The LLM prediction is evaluated at temperatures ranging from $T = 0.3$ to 1.

*Figure 25.* Noisy logistic map standard deviation $\sigma$ as a function of state value $X_t$, learned by the LLM, along with the ground truth. The LLM prediction is evaluated at temperatures ranging from $T = 0.3$ to 1.

## A.10. Hierarchy PDF

This section documents all three parts of the Hierarchy-PDF algorithm. We refer to the GitHub repository for further details.

---

**Algorithm 2** Refine Each State in a Stochastic Sequence

---

  **Input:**
  - $S_{\text{traj}}$: A string representing a sampled stochastic trajectory whose states are separated by commas.
  - $L_{\text{PDF}}$: List of unrefined PDFs for each state.
  - $KV_{\text{cache}}$: Key-value cache of running `model.forward`$(S_{\text{traj}})$.
  **for** each `state` and PDF in $S_{\text{traj}}$ and $L_{\text{PDF}}$ **do**
    PDF $\leftarrow$ RecursiveRefiner(True, `state`, $D_c, D_t, KV_{\text{cache}}$)
  **end for**

---

**Algorithm 3** Detailed Hierarchy-PDF Recursive Refiner

---

  **Input:** Object `multi_PDF` representing unrefined PDF using bins of various widths
  **Procedure:** RecursiveRefiner(mainBranch, sequence, $D_c, D_t, KV_{\text{cache}}$)
  **if** $D_c = D_t$ **then**
    **return** {Terminate if target refinement depth is reached}
  **end if**
  **if** mainBranch is True **then**
    {Launch 9 recursive branches if the current sequence is refined}
    $L_{\text{new}} \leftarrow$ Form 9 new sequences by changing the last digits
    **for** each `sequence` in $L_{\text{new}}$ **do**
      RecursiveRefiner(False, `sequence`, $D_c, D_t, KV_{\text{cache}}$)
    **end for**
  **else**
    {Collect refined logits}
    `newLogits, newKVcache` $\leftarrow$ NextTokenProbs(`sequence`, $KV_{\text{cache}}$)
    Refine `multi_PDF` using `newLogits`
  **end if**
  **if** $D_c + 1 < D_t$ **then**
    {Launch 10 more branches if $D_t$ not met}
    $L_{\text{new}} \leftarrow$ Form 10 new sequences by appending digits
    **for** each `sequence` in $L_{\text{new}}$ **do**
      RecursiveRefiner(False, `sequence`, $D_c + 1, D_t$, `newKVcache`)
    **end for**
  **end if**

---

**Algorithm 4** Extract Next Token Probabilities

---

  **function** NextTokenProbs(`sequence, KVcache, model`)
  `NextTokenLogit` $\leftarrow$ `model.forward`(`sequence, KVcache`)[last] {Extract distribution of next token}
  Update `KVcache`
  **return** `NextTokenLogit, KVcache`

---

