# OpenReview forum: "LLMs learn governing principles of dynamical systems, revealing an in-context neural scaling law"
_ICML.cc/2024/Workshop/ICL — ICML 2024 Workshop ICL Poster_

### Official Review · Reviewer_GxEA · 2024-06-07
**Great work, good fit for the workshop.**

**Rating:** 3
**Fit:** 3
**Confidence:** 3

**Workshop Review:**

This paper aims to show that pretrained LLMs are able to learn *in context* the transition rules of dynamical systems, and that it becomes more accurate as the length of the context increases. This ability is shown for various kind of dynamical systems (chaotic, stochastic, discrete, continuous, etc..).

As opposed to forecasting, this paper focus on how an LLM can understand the transition probabilities of a (markovian) physical phenomena underlying a given time series, with just a few timestamps observations. This makes it really different and novel to previous work on this field.

Strong points :

* No fine-tuning. Everything is done in context. No particular instruction-prompting.
* Highlighting a very interesting neural scaling law between the accuracy of the transition law predicted, and the context-length.
* Novelty of the proposed approach, and counter-intuitive results put forward by demonstrating the ability of language-trained models to understand randomly generated physical phenomena.

Weak points :

* The hierarchy-PDF algorithm is an interesting contribution, but it may be all the more interesting to adapt it to the vocabulary of newest LLMs. In fact, with newest models, the vocabulary size may grow and contains multi digits number tokens. For instance, GPT-4 and LLaMa-3 contain every $3$-digits numbers with their own token in their vocabulary, as opposed to LLaMa-2 used in the paper.
* The early-plateauing phenomena disscussed in Appendix A.8. seems to be a limitation of the method. It would be interesting to investigate more why the intuition provided (i.e. the fact that earlier data, are in some sense “out of distribution”) might be a correct explanation. In fact, discrete-state space markov chains of figure 12 seems to plateau (for $n \geq 9$), even if they have an invariant distribution, so it could be worth it to try to understand this beahvior deeply.
* Although this was mentioned in Appendix A.7., the fact that the LLMs used are trained to produce probability distributions of a token that depend only on the previous forces the method *for now* to limit itself to unigram Markov models (we can't beat the bi-gram baseline, figure 19). Maybe that's not such a problem for the future, given that people are already starting to think about initiatives to build LLMs that do multitoken predicition. [1]


References :

[1] Gloeckle, F., Idrissi, B. Y., Rozière, B., Lopez-Paz, D., & Synnaeve, G. (2024). Better & faster large language models via multi-token prediction. arXiv preprint arXiv:2404.19737.

Recommendation :

**Accept.**

**Reason For Not Giving Higher Score:**

N/A

**Reason For Not Giving Lower Score:**

Very complete, novel and clear work that meets the expectations of the workshop, in my opinion.

---

### Official Review · Reviewer_edXS · 2024-06-09
**This is interesting work and I enjoyed reading it**

**Rating:** 3
**Fit:** 3
**Confidence:** 2

**Workshop Review:**

The setup and methods are well-justified, and the results are compelling. Demonstrations of neural scaling laws in ICL are impactful in an era where context window sizes are increasing and potentially improving model capabilities. This is also a nice theoretical connection between in-weights learning and ICL. The domain of tokenized data generated by dynamical systems is simple but deep enough to effectively demonstrate ICL trends and compare LLM behavior to ground truth statistical models. In spite of being dense, I found this paper relatively easy to read and understand.

Additional notes:
- [1] shows an ICL scaling law with number of shots for few-shot jailbreaking LLMs.
- [2] uses a similar setup to yours, zero-shot prompting with varying ICL context lengths and comparing next-token probabilities to ground truth statistical models.
- Could the 10 state limit you observe with llama (Appendix A.3.3) be tied to single-digit tokenization? To my knowledge, llama models use single-digit tokenization [3]. It seems like this could be tested by prompting with comma-separated values as you use for continuous state spaces.
- Typo in Figures 13, 21, 22 "lenght"
- In Figure 3's histogram on the right, maybe I'm missing something here, but shouldn't the light blue shaded sections of the PDF (for the first step of the hierarchy) range between 500 and 599 since they're conditioned on the first digit being '5'?




[1] Many-shot Jailbreaking  https://www.anthropic.com/research/many-shot-jailbreaking

[2] In-Context Learning Dynamics with Random Binary Sequences  https://arxiv.org/abs/2310.17639

[3] Tokenization counts: the impact of tokenization on arithmetic in frontier LLMs  https://arxiv.org/abs/2402.14903v1

**Reason For Not Giving Higher Score:**

Prior work [1] not cited here also demonstrates ICL scaling laws

**Reason For Not Giving Lower Score:**

Setup and results are compelling. Figures are effective in explaining setup, and plots are effective in conveying key results.

---

### Meta-Review · Area_Chair_EZng · 2024-06-14

**Recommendation:** 2

**Metareview:**

The paper demonstrates how a transformer, primarily trained on text, can accurately predict dynamical system time series without fine-tuning or prompt engineering. Authors further show that the accuracy of the learned physical rules increases with the length of the input context window, revealing an in-context version of the neural scaling law. The results are interesting, and the paper is easy to understand and read.

All reviewers consider this paper favorably.

---

### Decision · Program_Chairs · 2024-06-17

Accept (Poster)